# Survival of Vaccine-Induced Human Milk SARS-CoV-2 IgG, IgA and SIgA Immunoglobulins across Simulated Human Infant Gastrointestinal Digestion

**DOI:** 10.3390/nu14163368

**Published:** 2022-08-17

**Authors:** Myrtani Pieri, Maria-Arsenia Maniori, Lucy Shahabian, Elie Kanaan, Irene Paphiti-Demetriou, Spyros Pipis, Kyriakos Felekkis, Vicky Nicolaidou, Christos Papaneophytou

**Affiliations:** 1Department of Life and Health Sciences, University of Nicosia, 46 Makedonitissas Avenue, Nicosia CY-1700, Cyprus; 2Cyprus Breastfeeding Association, Nicosia CY-2007, Cyprus; 3Medical School, University of Nicosia, Nicosia CY-2408, Cyprus; 4Aretaeio Hospital, 55-57, Andrea Avraamidi St, Strovolos, Nicosia CY-2024, Cyprus

**Keywords:** immunoglobulins, COVID-19, breastfeeding, vaccination, digestion

## Abstract

Breastfeeding can be a vital way of acquiring passive immunity via the transfer of antibodies from the mother to the breastfeeding infant. Recent evidence points to the fact that human milk contains immunoglobulins (Ig) against the SARS-CoV-2 virus, either after natural infection or vaccination, but whether these antibodies can resist enzymatic degradation during digestion in the infant gastrointestinal (GI) tract or indeed protect the consumers remains inconclusive. Herein, we evaluated the levels of IgG, IgA, and secretory IgA (SIgA) antibodies against the spike protein of SARS-CoV-2 in 43 lactating mothers who received at least two doses of either an mRNA-based vaccine (Pfizer/BioNTech, Moderna; n = 34) or an adenovirus-based vaccine (AstraZeneca; n = 9). We also accessed the potential persistence of SARS-CoV-2 IgA, IgG, and secretory IgA (SIgA) antibodies from vaccinated women in the GI tract of the infants by means of a static in vitro digestion protocol. Our data depict that, although slightly reduced, the IgA antibodies produced after vaccination resist both the gastric and intestinal phases of infant digestion, whereas the IgGs are more prone to degradation in both phases of digestion. Additionally, SIgA antibodies were found to greatly resist the gastric phase of digestion albeit showing some reduction during the intestinal phase. The evaluation of the vaccine induced Ig profile of breastmilk, and the extent to which these antibodies can resist digestion in the infant GI tract provide important information about the potential protective role of this form of passive immunity that could help decision making during the COVID-19 pandemic and beyond.

## 1. Introduction

The coronavirus disease 2019 (COVID-19) outbreak, caused by the severe acute respiratory syndrome coronavirus 2 (SARS-CoV-2), has posed global public health concern due to its high dispersion rate influencing public health, society, and the economy across the world [1,2]. To date, six vaccines have been approved to target COVID-19 by the European Medicines Agency (EMA) for use in Europe, two employing mRNA technology (BNT162b2 mRNA by Pfizer-BioNTech and mRNA-1273 by Moderna), two employing recombinant adenovirus technology (ChAdOx1 nCoV-19 by Oxford/AstraZeneca and Ad26.COV2.S by Johnson & Johnson) [3], one containing inactivated whole particles of the virus (Valneva) and one containing the spike protein of the virus (Nuvaxovid). Of all these, only two are authorized for use in children (Pfizer-BioNTech in children 5 years and older and Moderna in children 6 years and older). Vaccination in all children 6 months through 5 years of age has recently been recommended by the FDA in the USA using either the Pfizer/BioNTech or the Moderna vaccine. Even though young children are usually only mildly affected by COVID-19, there have been reports of the appearance of a multisystem inflammatory syndrome in this population, linked to SARS-CoV-2 infection that is both dangerous and potentially lethal [4,5,6,7].

For breast-fed infants, human milk is a source of various nutrients (e.g., proteins, peptides) and bio-active components that promote neonatal growth and protect from viral and bacterial infection [8]. In terms of antibodies, human milk contains an array of immunoglobulins (Igs), including IgA, secretory IgA (SIgA), IgM, secretory IgM (SIgM), and IgG (see [9] and references cited therein). IgA is the most abundant antibody isotype (~90% of the total Ig) in human milk [10,11], mostly found as SIgA; a dimer bound to a secretory component [12]. IgA antibodies in human milk protect the infant by eliminating invading pathogens on mucosal surfaces or systemically [11], while specific IgA antibodies have been detected in the human milk of mothers previously infected by SARS human immunodeficiency virus (HIV), and respiratory syncytial virus [11,13].

It has been previously demonstrated that human milk from mothers who have been infected with SARS-CoV-2 contains antibodies against the virus, and that these antibodies also demonstrate a neutralizing ability [14,15,16,17]. The same has been shown for mothers vaccinated with the mRNA vaccine technology, where SARS-CoV-2-specific IgG and IgA antibodies seem to appear in breastmilk soon after the first dose and reach maximum levels about 2 weeks after the second dose [16,18,19]. A number of pre-prints have also shown that, even though both IgA and IgG antibodies appear in milk after vaccination, IgG seems to be the prominent isotype reported in the majority of studies, but results are still inconclusive [18,19,20,21,22,23]. Additionally, the presence and persistence of specific SARS-CoV-2 antibodies in breast milk seem to be dependent on the vaccine type, with higher IgG and IgA levels in mRNA-based vaccines when compared to AstraZeneca, and on previous virus exposure [24].

Whichever the antibody profile secreted in milk upon vaccination, it is understood that in order for any such antibodies to play a role in immunoprotection, Igs must resist proteolytic degradation throughout digestion to remain intact and functional in order to bind pathogens in the digestive system and block that route of infection. Additionally, to provide a more systemic protection, these antibodies, must be absorbed into the bloodstream of infants [25]. Importantly, one study demonstrated that the vaccine-elicited Ig profile in milk after COVID-19 mRNA-based vaccination (Pfizer/BioNTech and Moderna) is IgG-dominant and lacks secretory antibodies, which is the Ab isotype class that is highly stable and resistant to enzymatic degradation in all mucosae [20]. To date, only one study examined whether the vaccine-induced antibodies withstand the infant digestion process with a small number of participants and found that these antibodies seem to resist digestion [26].

In this study, we first validated the accuracy of a commercially available ELISA assay for the determination of anti-SARS-CoV-2 IgAs and IgGs in breastmilk. We then employed these assays to evaluate the Ab response in breastmilk after vaccination with both doses of either the Pfizer/BioNTech, Moderna or the Astra Zeneca vaccine in 43 lactating participants. In 10 of these participants, we also evaluated the response following a booster dose. Finally, we assessed the persistence of anti-SARS-CoV-2 IgA, IgG, and SIgA antibodies following a static in vitro protocol of infant GI digestion. To our knowledge this is the largest study, to date, performed to explore whether SARS-CoV-2 antibodies withstand digestion.

## 2. Materials and Methods

### 2.1. Study Participants and Study Design

In the present study, 43 lactating women that had no diagnosis of laboratory-confirmed COVID-19 (with an antigen rapid test or RT-qPCR test) and were scheduled to receive a COVID-19 vaccine were recruited (n = 28 with BNT162b2 mRNA/BioNTech/Pfizer, n = 6 with mRNA-1273/Moderna and n = 9 with Vaxzevria/AstraZeneca). The vaccinated participants had received two or three doses of the mRNA vaccines. Control milk samples were obtained from the same individuals before vaccination and used as controls to establish positive cut-off values for each assay. Samples were also collected 1 day before the second dose, 3 weeks after the second dose, and, in some participants (n = 10), 3 weeks after the booster (3rd dose) dose to determine antibody levels. All samples were collected in sterile tubes, stored immediately at −20 °C and later divided into 1.5 mL aliquots and stored at −80 °C until used. On the collected breast milk samples, anti-SARS-CoV-2 IgA, IgG, and SIgA were evaluated prior and after simulation of infant GI digestion. The study design overview is shown in Figure 1.

In addition to breastmilk samples, clinical and demographic characteristics were also collected. Participants received information and gave written informed consent before enrolment. All procedures were approved by the Cyprus National Bioethics Committee (protocol number EEBK/EΠ/2021/24).

### 2.2. SARS-CoV-2 IgA and IgG ELISA Validation 

We previously optimized the use of commercially available ELISA kits specific for the determination of IgA and IgG immunoglobulins against SARS-CoV-2 in human plasma and serum (Abcam Cat numbers: ab277285 for IgG and ab277286 for IgA) [27]. Briefly, the spectrophotometric ELISA assays were monitored with a microplate reader (PerkinElmer, Waltham, MA, USA). The concentration of both IgA and IgG positive control samples (provided by the manufacturer) was determined using the Bradford method [28], while bovine serum albumin (BSA) was used to create a standard curve. Two human milk samples (500 μL each) were supplemented with 200 μg/mL IgA or IgG against SARS-CoV-2. The specific SARS-CoV-2 IgA and IgG ELISA assays were validated for accuracy and precision as previously described [29] with some modifications. All ELISA tests were performed for the two human milk samples on the same day with two dilutions and in triplicate. The accuracy of the assays was evaluated using the percentage error from the following equation: % Error = (V_A_ − Vo)/V_A_ × 100, where V_A_ is the known value and Vo represents the observed value. The precision of the assays was evaluated using the percentage coefficient of variation (CV), as follows: % CV = Standard deviation (SD)/Average × 100. The recovery of the assays was evaluated via the equation: % Recovery = (M_S_ − M_N_)/T_S_ × 100 [29], where M_S_ is the measured concentration of the spike sample, M_N_ is the measured concentration of the neat samples, and T_S_ is the theoretical concentration of the spiked sample. Unspiked milk was used as a blank in these experiments. The ELISA ab277285 kit was also utilized with modifications to evaluate the levels of SIgA, where the IgA antibody was replaced with the Sheep anti-Human secretory component (SHAHu/SC/PO, Nordic-MUbio) antibody in a dilution of 1:1000. This antibody reacts with both bound secretory component (secretory IgA) and with the free SC present in human secretions.

### 2.3. Determination of Anti-SARS-CoV-2 IgA, IgG, and SIgA Levels in Breast Milk

Anti-SARS-CoV-2 IgA, IgG, and SIgA were analyzed in the undigested breast milk samples and after the gastric and intestinal stages of simulated in vitro digestion. Milk samples were quickly thawed in a water bath at 37 °C and centrifuged at 1300× *g* for 20 min at 4 °C, fat was removed and skimmed fraction transferred to new tubes and centrifuged again under the same conditions to remove residual fat and cells. Fat-free milk was aliquoted and stored at −80 °C until testing. For the evaluation of the levels of IgA, IgG, and SIgA specific to SARS-CoV-2 spike protein in human milk, commercially available ELISA kits from Abcam were used with modifications as mentioned above. Milk was either undiluted or diluted 1:2 using assay buffer (provided by the manufacturer) (for IgA) or 1:2 (or 1:4) using assay buffer supplemented with 1% *w*/*v* BSA (for IgG) and added to the plate. Antibody levels were reported as Units (U) using the following equation (that was provided by the manufacturer) Units = ((Sample Absorbance × 10/Cut-off) × DF)), using, as cut-off, the mean absorbance values of milk samples before vaccination, where DF is the dilution factor. Samples containing > 10 Units of IgA or IgG were considered as positive. 

### 2.4. In Vitro Infant GI Digestion of Human Milk

A two-step in vitro infant GI digestion method that consists of a 60 min of gastric phase and a 120 min of intestinal phase, as previously described by [30,31], was used in this study with slight modifications. Briefly, for gastric digestion, milk samples (1 mL) were mixed with 500 μL of simulated gastric fluid consisting of 0.15 M NaCl, pH 4.0 supplemented with 0.3 mg pepsin (22.75 U/mg total protein). The pH of each sample was re-adjusted to 4.0 ± 0.05 with 1 N HCl. The samples were then incubated at 37 °C with continuous shaking at 300 rpm for 60 min. When the incubation time for the gastric phase of digestion elapsed, a 300 μL aliquot was removed and stored at −80 °C. The remaining samples were further processed for the simulated intestinal digestion. The pH of each sample was adjusted to 8.0 by adding 1 N NaOH. Bile salts (0.4 mg) (Sigma-Aldrich, St. Louis, MO, USA) were added to make a final concentration of 2 mM. Porcine pancreatin (0.1 mg; 3.45 U of pancreatin/mg of total protein) (8 × USP, Sigma-Aldrich) was added to the mixture and the pH was readjusted to pH 8.0 ± 0.05. The mixture was incubated at 37 °C with shaking at 300 rpm for 120 min. At the end of the incubation, samples were stored at −80 °C until further analysis.

### 2.5. Statistical Analysis 

Unless otherwise stated, data are reported as means ± standard deviation (SD). The normal distribution of continuous data was analyzed with the D’Agostino and Pearson omnibus normality test. Antibody levels were log-transformed before all statistical processes. Log values of antibody levels follow a normal distribution and, thus, are presented as mean ± SD.

The levels of antibodies before and after vaccination were compared via one-way ANOVA with Tukey’s multiple comparison test was used to compare percentage stability of each antibody before and after digestion. All reported *p*-values were two-tailed and *p*-values < 0.05 were considered statistically significant. Statistical analysis was performed using GraphPad Prism (v.8.0, GraphPad Software Inc., San Diego, CA, USA).

## 3. Results

### 3.1. Subject Characteristics

Characteristics of study participants and their infants are presented in Table 1. 

### 3.2. SARS-CoV-2 IgG and IgA ELISA Method Validation 

We have preciously validated two commercially available ELISA kits from Abcam (developed for use with serum and plasma samples), to evaluate the levels of SARS-CoV-2-specific IgA and IgG in human milk samples [27]. Initially, the ELISA assays were validated using IgA and IgG spiked human milk samples (Table 2). In detail, two human milk samples spiked with either IgA or IgG specific to SARS-CoV-2 were analyzed within a single day with two dilutions for each sample and three replicates. The SARS-CoV-2 IgA and IgG ELISA assays were highly accurate with 5.17% and 18.25% error, respectively. The % CV (co-efficient variation) of SARS-CoV-2 IgA and IgG ELISA assays were 8.51 and 16.73%, respectively, which meets the minimal validation requirements for assay precision (i.e., % CV < 30) according to [29]. It should be noted that as previously shown, the addition of BSA to a final concentration of 10 mg/mL significantly increased (*p* < 0.001) the recovery of spiked-IgG in human milk samples from 53.6% to 83.75% (data not shown) and decrease the assay error from ~50% to 18.25% [27]. Therefore, BSA was included in the subsequent ELISA experiments for the determination of IgG levels in human milk samples. On the contrary, the presence of BSA did not affect spike IgA recovery and assay percentage error [27].

### 3.3. Evaluation of Anti-SARS-CoV-2 IgA and IgG Levels in Milk Samples after Vaccination

We next evaluated the presence of IgG and IgA antibodies against SARS-CoV-2 in breastmilk of 43 participants after vaccination with two doses. Of those participants, 10 (23.3%) also donated a sample after the booster dose, six months after the second vaccine dose, therefore IgG and IgA antibodies were evaluated in those samples as well. Samples were collected one day before the first dose, one day before the second dose and three weeks after the second and booster dose (Figure 1). Fat-free breastmilk samples were assayed in separate commercially available ELISAs measuring either IgA or IgG against SARS-CoV-2 (Figure 2). Samples were defined as IgA or IgG positive if the measured Ig units were greater than 10, the pre-vaccination (pre-vac) samples were used as a control (see methods, Section 2.3 Determination of IgA and IgG levels). 

Significantly higher antibody responses were found in breastmilk samples collected after the administration of the second vaccine dose, as compared to baseline, for both the IgG (*p* < 0.05) and IgA antibodies (*p* < 0.001) (Figure 2). A further, albeit not significant, increase was seen in the samples collected three weeks after the booster dose as compared to the second vaccine dose. 

### 3.4. Persistence of Anti-SARS-CoV-2 IgAs, IgGs and SIgAs Following the Gastric and Intestinal Phases of In Vitro GI Digestion

Of our cohort of 43 lactating participants, 35 of them had donated enough milk in order to further perform digestion analysis. Therefore, 35 samples were subjected to simulated infant GI digestion and concentrations of anti-SARS-CoV-2 IgA, IgG and SIgA were determined using the ELISA assays, as described previously. Vaccine-induced anti-SARS-CoV-2 Ig levels were decreased through the gastric phase and further through the intestinal phase of digestion as compared to basal levels (Figure 3). However, antibodies were found to persist in the samples whilst different persistence patterns were observed for the different Ig isotypes. Generally, IgA antibodies were shown to better withstand digestion with mean values remaining well above the cut-off values. SIgAs were stable especially following the gastric phase after which only minimal decrease was observed. On the contrary, vaccine-elicited anti-SARS-CoV-2 IgGs were degraded in both gastric and intestinal digestion steps. Specifically, the gastric phase of digestion resulted in 17.1%, 8.6%, and 3.2% degradation for the IgG, IgA, and sIgA antibodies, respectively. The intestinal phase of digestion resulted in 74.3%, 14.3%, and 48.4% of degradation for the IgG, IgA, and sIgA antibodies, respectively (Figure 3).

## 4. Discussion

In the present study, we demonstrated that commercially available ELISA kits specific for the detection of anti-SARS-CoV-2 IgG, IgA and SIgA in human serum and plasma can be utilized with some modifications for the detection of these Igs in human breastmilk samples. All three assays were highly accurate and met the typical validation requirements for their intended use. 

Subsequently, using these assays we detected anti-SARS-CoV-2 antibodies in breastmilk in a Cyprus cohort of 43 lactating participants that were vaccinated with the Pfizer/BioNTech, Moderna or AstraZeneca vaccines. It was shown that anti-SARS-CoV-2 antibodies could be detected in the breastmilk of vaccinated individuals with levels further increasing following the second dose of vaccination. Our findings agree with several other studies that have demonstrated the induction of anti-SARS-CoV-2 antibodies in the breastmilk following vaccination. In this study, we were also able to investigate several samples from 10 individuals who had received a booster vaccination, and showed that antibody levels were further increased, supporting that booster vaccination is useful not only for continuous protection of the individual but potentially also of the breastfeeding infant. To our knowledge, only two other studies have evaluated the presence of SARS-CoV-2 antibodies in breastmilk after a booster dose: a study of four individuals conducted by Leung et al., which also showed that booster dose is able to re-trigger the secretion of neutralizing IgA in breastmilk [32], and a preprint study of 12 participants also depicting increase in SARS-CoV-2 RBD-specific IgA and IgG antibodies in breast milk from lactating women following the COVID-19 booster vaccination [33].

Several studies have shown that Igs against SARS-CoV-2 in human milk induced after infection or vaccination have a neutralizing ability, suggesting their potentially protective role [20,34,35,36,37]. However, this protective role, i.e., neutralizing pathogens on mucosal surfaces or in the infant gut, will heavily depend on the ability of the antibodies to withstand degradation in the consumer GI tract to their site of action. The stability of Igs in digestion has been studied and it has been previously demonstrated that gastric digestion may reduce IgGs, but other antibodies, including IgAs, are not digested in the gastric contents of preterm infants [38]. Additionally, a previous study revealed a greater reduction in IgAs compared to IgGs or IgMs in preterm infant stools [39]. Two oral supplementation studies (in adults fed bovine colostrum SIgA/IgA, IgM, and IgG, and in preterm infants fed serum IgA and IgG [39,40]) demonstrated that IgG and IgM survive intact to the stool, whereas SIgA/IgA does not. However, some studies have demonstrated that human milk-derived SIgA survived intact to the infant stool and urine [41,42,43]. It has also been shown that, overall, the stability of human milk Igs during gastric digestion is higher in a preterm infant than in term infants [38]. In infants, children, and adults, the amount of intact IgG recovered in stool ranges from trace amounts up to 25% of the original amount ingested [44]. 

Our previous preliminary study was the first to investigate the persistence of anti-SARS-CoV-2 antibodies in breastmilk from vaccinated lactating women by means of an in vitro simulated infant GI digestion [27]. In the current study, we used a larger cohort of lactating women to verify previous findings and present new ones. More specifically, 35 samples were subjected to gastric and intestinal digestion and levels of IgG, IgA, and SIgA were evaluated. Although antibody levels decrease after gastric digestion, they are still detectable in the majority of samples. Additionally, different persistence patterns were observed depending on the Ig isotype, with IgA and especially SIgA being more robust, since only minimal decrease was seen following gastric digestion. Intestinal digestion further decreased the Igs levels, but in many cases, these were still detectable. These findings suggest that these antibodies can be immunologically active, and, hence, protective, in the baby’s lumen during breastfeeding. 

The study by Calvo-Lerma et al., is the only other study that has investigated the survival of IgA and IgG specific for SARS-CoV-2 after in vitro simulated digestion [26]. In this study it is also shown that antibodies can survive digestion, with levels remaining above cut-off values, even following the intestinal phase of digestion. The authors suggest that the increased survival of the antibodies, especially in the intestinal phase, may be attributed to differences in the simulated conditions, including a different pH, different enzymatic concentrations, and less duration. For example, in Calvo-Lerma et al., the intestinal phase samples were subjected to digestion for 60 min whereas in our study for 120 min; such differences will of course influence the outcome. This illustrates the limitations of such simulated protocols and difficulties in comparing findings from different studies. Nevertheless, this does not negate the fact the antibodies do persist and can be active until their inevitable eventual degradation, especially in exclusively breastfed infants. 

An important strength of this study was the evaluation of SIgAs in breastmilk samples of vaccinated individuals. The majority of IgAs in human milk are SIgAs, which have a primary role as a protective agent in human milk, protecting mucosal surfaces against various pathogens, including viruses, by inhibiting the pathogen adherence to these surfaces [38]. Whether the intramuscular vaccination (IM) elicits the production of SIgA remains inconclusive. Studies using non-human primates (NHPs) revealed that an IM vaccine may not promote the production of SIgAs [45], whilst in a cohort of 50 lactating individuals less than 50% of milk exhibited specific SIgA, with mRNA-based vaccine recipients SIgA titers measuring significantly higher than those of adenoviral-based vaccines [46]. In our study, we have found that anti-SARS-CoV-2 SIgA are induced in breastmilk following vaccination. 

SIgA is found as dimers containing the highly glycosylated secretory component and J chain, which make them less susceptible to proteolysis. This may explain our findings that following gastric digestion there is only minimal decrease in sIgA levels. This is indeed the first study that has differentiated the anti-SARS-CoV-2 SIgAs present in breastmilk and has investigated their resilience through digestion. Nevertheless, further studies are required to show whether the digested breastmilk has the potential to neutralize the virus, something that has not been studied to date. 

In this study, we did not include individuals that had been infected with SARS-CoV-2, therefore conclusions regarding differences in various antibody isotype levels cannot be drawn. Additionally, this study did not measure IgA and IgG concentrations in the serum of the lactating mothers, thus comparisons of antibody titers between serum and milk could not be performed. However, it has been reported that the antibody response to COVID-19 vaccination was rapid and highly synchronized between breastmilk and serum of lactating mothers [47]. Further studies are also needed to elucidate the correlation between IgA/SIgA and IgG levels after vaccination with the age of infant and duration of lactation, as well as the nature of vaccine-elicited Igs based on the type of the vaccine; we have not done this type of analysis here. It has been proposed that the immunological profile of mother’s milk is dynamic and can be influenced by several factors including the week of gestation and lactation period [48]. Therefore, the age of the infant and the duration of lactation should also be taken into account when studying Ig profile in human milk post-vaccination. We performed an analysis of our data based on vaccine type and were able to demonstrate a trend towards higher milk secretion of IgG antibodies in the mRNA vaccines vs. the adenoviral vaccine, albeit significant only after the first dose. Still, this trend agrees with current literature depicting that mRNA-based vaccines resulted in higher SARS-CoV-2 antibody responses in human milk compared to vector-based vaccines [49].

This study represents the biggest study up to now in Europe reporting specific antibody levels in milk for three different vaccines. Our study provides preliminary data, and it has limitations. Larger prospective studies, bigger samples sizes, and distinct cohorts in different locations and long-term follow-up are needed to better understand the time course of SARS-CoV-2 immunity in milk and the possible differences in the presence of antibodies in breast milk according to vaccine type. In addition, longer follow-up of the antibody levels would be needed in order to determine their persistence. Furthermore, although it has been suggested that antibodies found in breastmilk would exert strong neutralizing effects, no functional assays were performed. In addition, the potential impact on neonatal growth as well as the potential protective effect against infection in the infant remains elusive.

## 5. Conclusion

In this study, anti-SARS-CoV-2 antibodies were detected in breastmilk samples of a Cyprus cohort of 43 lactating participants that were vaccinated with the Pfizer/BioNTech, Moderna or AstraZeneca vaccines. Antibody levels in milk were shown to be increased following the second dose and booster doses of vaccination. Notably, this is also the biggest study to date to examine the resilience of IgG, IgA and SIgA antibodies via digestion and to show that a number of SARS-CoV-2 antibodies can withstand digestion, with levels remaining above cut-off values, even following the intestinal phase of digestion. Overall, these findings suggest that vaccine-induced anti-SARS-CoV-2 antibodies can be immunologically active throughout infant digestion, and, hence, protective, in the baby’s lumen during breastfeeding. Further studies will confirm these findings.

## Figures and Tables

**Figure 1 nutrients-14-03368-f001:**
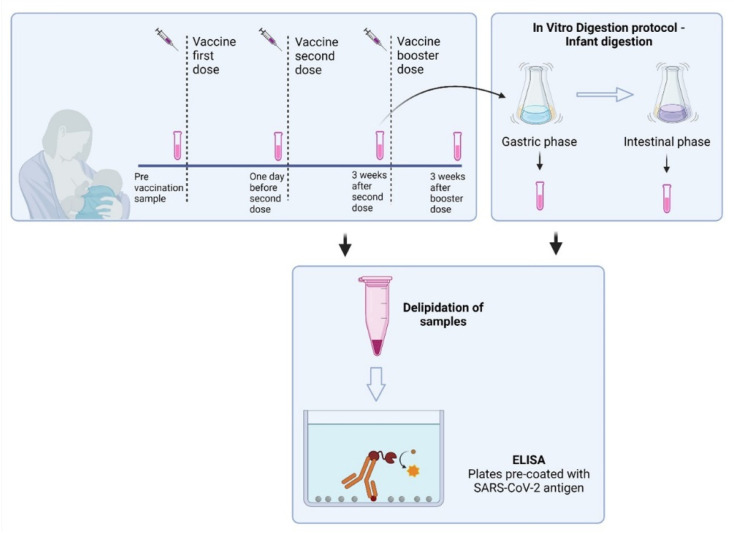
Overview of the study design.

**Figure 2 nutrients-14-03368-f002:**
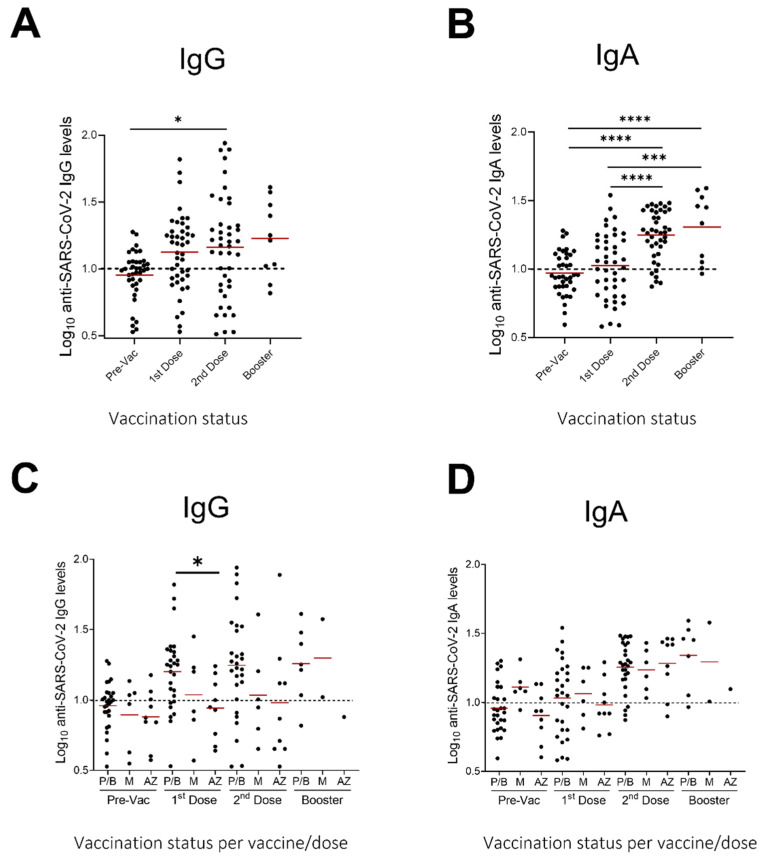
Levels (in relative Units) of anti-SARS-CoV-2 IgG and IgA in human milk samples. SARS-CoV-2 IgG (**A**) and IgA (**B**) antibodies before and after 1st, 2nd, and booster dose of vaccination (all vaccines) and SARS-CoV-2 IgG (**C**) and IgA (**D**) antibodies before and after 1st, 2nd, and booster dose of vaccination (per vaccine type) Dotted lines: positive cut-off values previously determined for each assay as the mean values of negative control (before vaccination) milk samples. Asterisks show statistically significant differences between variables; (* *p* < 0.05; *** *p* < 0.001; **** *p* < 0.0001) using one-way ANOVA followed by Tukey’s multiple comparisons test. P/B: Pfizer/BioNTech, M: Moderna, A/Z: AstraZeneca.

**Figure 3 nutrients-14-03368-f003:**
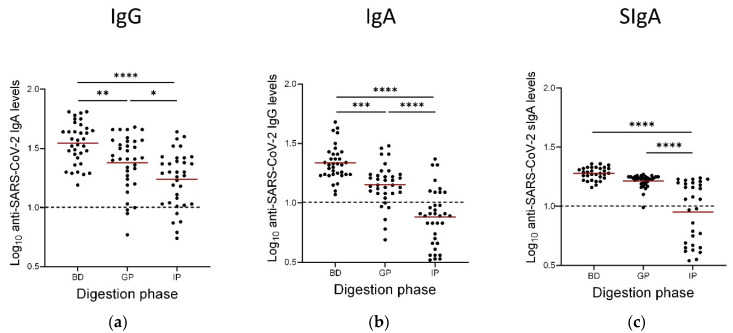
Concentrations of IgG (**a**), IgA (**b**), and Secretory IgA (SIgA) SARS-CoV-2 antibodies (**c**) in milk before and after a 1 h simulated gastric digestion and a 2 h simulated intestinal infant digestion protocol. Dotted lines: positive cut-off values previously determined for each assay as the mean values of negative control (before vaccination) milk samples. Asterisks show statistically significant differences between variables (* *p* < 0.05; ** *p* < 0.01; *** *p* < 0.001, **** *p* < 0.0001) using one-way ANOVA followed by Tukey’s multiple comparisons test. BD: before digestion, this sample was taken 3 weeks after the second vaccine dose, GP: After the Gastric phase of digestion, IP: after the intestinal phase of digestion. Horizontal bars indicate mean values.

**Table 1 nutrients-14-03368-t001:** Characteristics of the participants included in the study.

Vaccine	BioNtech/Pfizer n = 28	Moderna n = 6	AstraZeneca n = 9	Total n = 43
Age (years)	36 ± 3.7	37 ± 2.3	37 ± 3.1	36 ± 3.4
Maternal co-morbidities				
Chronic hypertension	0% (0 of 28)	0% (0 of 6)	0% (0 of 9)	0% (0 of 43)
Diabetes/gestational Diabetes	0% (0 of 28)	0% (0 of 6)	0% (0 of 9)	0% (0 of 43)
BMI (kg/m^2^) > 30	3.6% (1 of 28)	0% (0 of 6)	0% (0 of 9)	2.3% (1 of 43)
Asthma	2% (0 of 28)	0% (0 of 6)	1% (0 of 9)	7.0% (3 of 43)
Immunosuppression/cancer	10.7% (3 of 28)	0% (0 of 6)	22.2% (2 of 9)	11.6% (5 of 43)
Gestational age (weeks)	84 ± 47.9 ^1^	82 ± 35.9	81 ± 19.7	83 ± 40.3
Timing between two doses (days)	21	28	28–84	
Timing of booster dose (months after 2^nd^ dose)	6	6	6	
Participants that provided milk after booster dose	7	2	1	10

^1^ Data are presented as mean values ± Standard Deviation.

**Table 2 nutrients-14-03368-t002:** SARS-CoV-2 IgA and IgG ELISA method validation.

Parameter	IgA	IgG	sIgA
Error (%)	5.17	18.25	6.21
Precision (% CV)	8.51	16.73	10.14

## Data Availability

Not applicable.

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
