# Peer review of "Survival of Vaccine-Induced Human Milk SARS-CoV-2 IgG, IgA and SIgA Immunoglobulins across Simulated Human Infant Gastrointestinal Digestion"

_nutrients, 2022, doi:10.3390/nu14163368_

Round 1

Reviewer 1 Report

Overall, a very nice study with interesting results. I have a few comments which I believe need addressing:

-Table 1: would be good to include timing of booster as well

-Table 2 should be removed in my opinion. This is outside the scope of the study and numbers are far too small to give any reliable outcomes for these self-reported measures.

-Figure 2 results: are there any trends to be found here with respect to vaccine type? Would be interesting to know

-Figure 3: legend should include anti-SARS-CoV 2 for the Ig’s?

Author Response

Point 1: -Table 1: would be good to include timing of booster as well

Response 1: Thank you. Booster timing has been added to Table 1 and also in line 222 of the revised manusript.

Point 2: -Table 2 should be removed in my opinion. This is outside the scope of the study and numbers are far too small to give any reliable outcomes for these self-reported measures.

Response 2: We have removed Table 2. Indeed our numbers are small to make any solid conclusions.

Point 3: -Figure 2 results: are there any trends to be found here with respect to vaccine type? Would be interesting to know

Response 3: We have now done an analysis with respect to vaccine type and we are showing the results in figure 2 (C,D). There is a trend towards higher milk antibody detection in the mRNA vaccines as compared to the adenoviral vaccine, albeit only significant after the 1st dose between the Psizer/BioNTech and the Astrazeneca vaccines. We have also added a sentence in the dicussion section (lines 371-376) about the trend observed.

Point 4: -Figure 3: legend should include anti-SARS-CoV 2 for the Ig’s?

Response 4: Thank you. This had been corrected.

Reviewer 2 Report

Study design is very clear and accurate. The only major concerns about the research are correctly stated in the limitations section.

I only have a comment regarding table 2. I do not see the relationship of the table (side-effects after vaccination) with the topic of the article.

The data in this table is not used in the further part of the study or included in the discussion. I believe that the table is unnecessary and I suggest deleting it.

Author Response

Point 1: -I only have a comment regarding table 2. I do not see the relationship of the table (side-effects after vaccination) with the topic of the article.

The data in this table is not used in the further part of the study or included in the discussion. I believe that the table is unnecessary and I suggest deleting it.

Response 1: Thank you. We have now removed Table 2 from the revised manuscript.